# The Paradox of Group Citizenship and Constructive Deviance: A Resolution of Environmental Dynamism and Moral Justification

**DOI:** 10.3390/ijerph17228371

**Published:** 2020-11-12

**Authors:** Tingting Liu, Yahui Chen, Chenhong Hu, Xiao Yuan, Chang-E Liu, Wei He

**Affiliations:** 1College of Business Administration, Hunan University of Technology and Business, Changsha 410205, China; 70107@hutb.edu.cn (T.L.); 201910310082@stu.hutb.edu.cn (C.H.); 201810310029@stu.hutb.edu.cn (X.Y.); 2School of Business and Tourism Management, Yunnan University, Kunming 650106, China; 22018000068@mail.ynu.edu.cn; 3Mobile E-Business Collaborative Innovation Center of Hunan Province, Key Laboratory of Hunan Province for Mobile Business Intelligence, College of Business Administration, Hunan University of Technology and Business, Changsha 410205, China; 4Scott College of Business, Indiana State University, Terre Haute, IN 47802, USA

**Keywords:** constructive deviance, group citizenship behavior, environmental dynamism, moral justification, moral licensing theory

## Abstract

Previous research on antecedents to constructive deviance remains scattered and inclusive. Our study conceptualizes constructive deviance from the perspective of ethical decision making and explores its antecedents, mechanism, and conditions. Drawing on moral licensing theory and social information processing theory, we propose that group citizenship behavior facilitates moral justification and constructive deviance when environmental dynamism is high and inhibits them when it is low; and moral justification fully mediates the relationship between the interaction of group citizenship behavior and environmental dynamism and constructive deviance. With two-wave panel data collected from 339 employees in 54 groups of five service companies in retailing, finance, and tourism randomly selected from three provinces in southern China, these hypotheses are all supported empirically. Our findings broaden the antecedents and occurrence mechanism of constructive deviance through an ethical decision-making lens. Our study contributes to the moral licensing literature by enriching the sources of moral licensing in the workplace and empirically demonstrating that moral justification may function as an underlying mechanism of moral licensing.

## 1. Introduction

In a turbulent time when organizations increasingly face a dynamic and uncertain environment, employees’ agility and resilience become more essential to organization success than ever before. Organization leaders and managers have realized that employees’ failure to follow organizational norms and rules are not necessarily a bad thing all the time—sometimes, it can benefit their organization’s innovation and competitiveness instead [1,2]. Researchers have found that employees do engage in a variety of behaviors in the workplace that can be seen as both deviant and positive [2]. They call this type of behavior that violates the norms of a referent group to help the organization “constructive deviance” [2,3,4]. A variety of studies have confirmed that constructive deviance can help organizations respond to volatile environment and promote organizational creativity and change [2,3,5]. For example, supermarket sales associates who had been allowed to adopt better ways of serving customers by deviating from common ways of working (such as providing immediate discounts) were found to be able to help reduce loss to the company in terms of curbing waste or shrinkage [5]. As these studies showed, constructive deviance can build a powerful basis for organizational learning and improvement [5].

However, empirical research on antecedents of constructive deviance overall remains scattered and inconclusive [3,6]. Given the ethical nature of the construct [7,8], our study focuses on one important antecedent of constructive deviance through the lens of ethical decision making—how group factors influence the emergence of constructive deviance. In particular, constructive deviance is rife with ethical concerns, such as whether the decision to depart from the norms of a referent group to maximize organizational benefits is warranted at the expense of violating organizational obligations [8,9]. For that reason, the decision to engage in constructive deviance, even with good intentions, often appears to be a difficult ethical dilemma [8,9]. Moreover, few studies explored the antecedents and occurrence mechanisms of constructive deviance from the ethical perspective [7,8,9].

We believe moral licensing theory can be an effective explanation to the process in which people make ethical decisions when facing the dilemma of constructive deviance [10,11]. The theory suggests that, when confronted with the prospect of engaging in suspect actions, employees build confidence from their past good deeds [10,11]. Moreover, studies along this line found that one’s group members’ moral choices, reputation, and citizenship behavior can also become the sources of an individual’s moral licensing [12,13,14]. Some researchers also called for research broadening the source of moral licensing at the group level [15]. Therefore, we propose that group citizenship behavior, defined as “a group-level phenomenon concerning the extent to which work groups engage in behaviors that support other workgroups and the organization as a whole” ([16], p. 275), can elicit vicarious moral licensing among employees and influence their ethical decision making on constructive deviance. Since the pro-organization motivation of constructive deviance is implicit, it could be misconstrued as an ethical violation unless its true drive can be detected by others [6,17].

Moreover, previous research has revealed that individuals might form different understandings of constructive deviance with diverse environmental dynamism—particularly, when employees are in a highly dynamic environment, the implicit pro-organization motivation of constructive deviance can be easily detected by their organization [18,19]; group citizenship behavior thus can enhance the employees’ moral self-concept, provide a psychological environment for judging the morality of constructive deviance, justify constructive deviance, and ultimately, promote constructive deviance. Therefore, our study draws on moral licensing theory [11,20] and social information processing theory [21] to predict when and how group citizenship behavior facilitates employees’ constructive deviance. To be specific, we suggest that group citizenship behavior and constructive deviance have a positive correlation only under certain conditions, such as in the context of high environmental dynamism. Furthermore, we explore the mediation effect of moral justification on the relationship between the interaction of group citizenship and environmental dynamism and constructive deviance. That is, when the organization is in a dynamic environment, group citizenship behavior justifies constructive deviance as a pro-organizational act and thus, facilitates the occurrence of constructive deviance, since employees can omit the threat of damage to their moral self-image and organizational punishment.

Our study can make three contributions to the existing management literature on constructive deviance and moral licensing research. First, previous research on constructive deviance mostly adopted social exchange theory [22] or social information processing theory [23] to explore the antecedents of constructive deviance. Considering the ethical nature of constructive deviance, our work draws on moral licensing theory to uncover the conditions of constructive deviance in the context of ethical decision making. Second, our study establishes a theoretical and empirical link between group citizenship behavior and constructive deviance. It therefore contributes to the behavioral ethics literature through demonstrating the role of environmental dynamism in the moral licensing process and providing support for the underlying mechanism of the moral licensing process (i.e., moral justification). Last but not least, our study expands the moral licensing literature by revealing that group behavior can be an important source of moral licensing and moral justification.

## 2. Literature Review and Hypotheses

### 2.1. Constructive Deviance, Environmental Dynamism, and Group Citizenship Behavior

Constructive deviance is a special type of deviant behavior that can improve and contribute to the wellbeing of an organization and its members [3,4]. By nature, constructive deviance has dual attributes [3]: on the behavioral intention side, it is ethical since it is intended to be beneficial to the organization [7,8]; on the action side, however, it is unethical since it violates organizational rules or norms, challenges the established administration system, and causes management chaos [7,8].

Social information processing theory [21] proposes that people receive cues from the social environment surrounding them and form perceptions of the action that they intend to take. In other words, individuals can learn their behaviors by recognizing the informational and social environment where the behavior occurs [21]. Specific to the case of constructive deviance, people will form different understandings about it, subjective to various social cues—situational factors play a remarkable role in the understanding of constructive deviance [18].

Research further pointed out that one such social factor—environmental dynamism—can significantly affect individuals’ cognition of constructive deviance [18,19]. Environmental dynamism refers to the frequency of change and the degree of instability of the environment faced by organizations or groups [24]. It should be especially noted that environmental dynamism is widely explored at the team/group level as well as at the organizational level [25,26], especially in the era of team as the main unit of innovation. According to the perspective of social information processing, employees may have diverse views on constructive deviance under different degrees of environmental dynamism. Specifically, a group in a highly dynamic environment demands agility and adaptability of the employees to make sure that the group is responding to emerging changes promptly [25]. In such a situation, the positive aspect of constructive deviance, therefore, is easy to be aware of by others, since it well responds to group call for agility. Hence, when the group environment is highly dynamic, constructive deviance is more likely to be perceived as ethically ambiguous due to its dual attributes. On the contrary, a group operating in a more stable environment faces high predictability in job objectives and contents. In this situation, people engaging in constructive deviance can be regarded as “black sheep” because their pro-organization motivation cannot be easily spotted. Thus, when environmental dynamism is low, constructive deviance could be mistaken for destructive deviance by others.

Meanwhile, moral licensing theory states that individuals who previously engaged in socially desirable conducts might indulge themselves with morally questionable acts later [10,11]. While much is already known about individuals’ own morality and their intemperance, other research has probed other sources of moral licensing, such as other people’s moral acts, group loyalty and reputation, and so forth [12,13,14,27]. Findings from numerous studies demonstrated that people could incorporate attributes relevant to a group member behavior into their own self-concept and subsequently, adjust their own behavior accordingly [12,13]. Researchers named this special moral licensing phenomenon the “vicarious licensing effect” [12,13], which demonstrates that individuals can experience the positive moral behavior of others as their own and are likely to show a licensing effect akin to those demonstrated on others [12,13]. In other words, people can acquire moral licensing by observing the moral behavior of their reference group such as group citizenship behavior. In addition, research has suggested that the premise of the vicarious licensing effect is attributional ambiguity (i.e., multiple possible explanations) for the target behavior [12,28]. That is, others’ good deeds change the meaning of subsequent behaviors [12]. More specifically, others’ behavior or group behavior serves as a lens through which one can construe current behavior; when the motivation for current behavior is ambiguous, it is disambiguated in line with past behavior. For example, when two reasons compete (e.g., favoring White applicants due to racism versus maintaining a hostile working environment to Black people), group moral behavior can invalidate the morally illegitimate one, hence one can have confidence that the legitimate explanation will prevail [12]. Taken together, we can conclude that the vicarious licensing effect can only prevail when the target behavior is morally dubious [10,28].

Group citizenship behavior is how the group as a whole help other workgroups or the organization, rather than an aggregate of the ratings of members’ citizenship behavior [16]. Scholars have theorized that group-level citizenship behavior can influence employees’ behavior by fostering group-level norms [16]. Thus, combining vicarious moral licensing theory with the group citizenship literature, we can speculate that group citizenship behavior can trigger employees’ constructive deviance when the environment is dynamic. By contrast, as discussed before, constructive deviance is an overt violation when environmental dynamism is low, which can invalidate the vicarious licensing effect. Furthermore, in this case, group citizenship behavior creates behavioral norms, which can inhibit employees from performing constructive deviance that could be misunderstood as destructive deviance. Thus, we put forward the following hypothesis:

**Hypothesis** **1.**
*Group citizenship behavior and environmental dynamism interacts to influence constructive deviance, such that the relationship between group citizenship behavior and constructive deviance will be positive when the environmental dynamism is high and negative when it is low.*


### 2.2. Moral Justification and Constructive Deviance

Moral justification refers to the process that individuals rationalize their understanding of behaviors [27]. Research suggests that moral justification provides reasons for questionable behaviors and makes them appear less unethical or morally praiseworthy [29]. Therefore, moral justification attenuates the threat to one’s moral self-image when one engages in morally questionable behaviors [25,30]. According to the aforementioned discussion, the moral licensing mechanism describes the process that good deeds change the meaning of subsequent behaviors. Namely, when two explanations about the current behavior compete against each other (e.g., hiring a male staffer because of sexism versus because of his competence), past behavior (e.g., “I have shown I am not a sexist”) can invalidate the morally illegitimate one [10].

Following this logic, our study proposes that moral justification can act as a mediator between the interaction of group citizenship behavior and environmental dynamism and constructive deviance. Empirical research has showed that a psychological connection between two individuals can cause one to vicariously justify the other’s decisions [30]. Similarly, numerous studies have documented that group represents a compelling force that can be mobilized to justify individuals’ vicious actions [12,27,31]. More specially, group past moral action establishes in-group non-prejudiced credentials and then, justifies their suspected discrimination behavior [12]. Meanwhile, group citizenship behavior facilitates group loyalty [32], which is a fundamental facilitator of moral justification [27]. Furthermore, according to moral licensing theory, people use pro-violation justifications to redefine morally ambiguous behaviors as “non”-violations. Ambiguity and gray areas allow people to blur the difference between moral and immoral, and diminish the threat to their moral selves [29]. According to this logic, moral justification can accelerate constructive deviance. Taken together, we speculate that group citizenship behavior can promote group members’ constructive deviance through reframing the questionable behavior as acceptable and eliminating the guilt or anxiety of being punished. This reframing and elimination process is the mediation effect of moral justification [29]. Thus, combining moral licensing theory and social information processing theory, we suggest that, when environmental dynamism is high and constructive deviance is ambiguous, it is more likely for in-group individual employees to justify their nonconforming behaviors as a necessity for organizational competitiveness. By contrast, when the group environment is conventional and predictable, if constructive deviance blatantly challenges organizational rules or norms, such behavior cannot be justified and forgiven. Similarly, group citizenship behavior can foster group-level organizational citizenship behavior norms [16]. Under these circumstances, employees are afraid to be excluded from the group and be punished if they engage in such “destructive” deviance [33]. Therefore, employees are reluctant to perform constructive deviance due to their concerns about group citizenship behavior. Based on the discussion, we therefore hypothesize:

**Hypothesis** **2.**
*Moral justification mediates the interaction effect between group citizenship behavior and environmental dynamism on constructive deviance, such that the indirect effect will be positive when environmental dynamism is high and negative when it is low.*


Combining the two hypotheses together, we summarize our research variables and hypotheses in a conceptual framework in Figure 1.

## 3. Method

### 3.1. Participants and Procedures

After our research proposal had been approved by the academic ethics committees of our institutions (IRB 868-603), we collected data from full-time employees in five service companies in retailing, finance, and tourism located in three provinces in southern China. We purposely chose our participants from such groups as sales, customer services, and research and development in those types of service companies for two reasons—these jobs need more flexibility and are more often engaged in constructive deviance [34] and the scale on constructive deviance that we used in this research (see below for details) includes customer service items [4].

In order to control potential common variance bias, we conducted the survey in two phases since temporal separation between the measurement of independent and dependent variables can avoid social desirability and prevent reverse causality [35]. The questionnaire for the first phase contained items on group citizenship behavior, group identification, environmental dynamism, and demographic information (see below for details on these measures). The second phase (three weeks later) questionnaire asked the participants about their moral justification and constructive deviance. With the assistance of their human resource departments, we paired the participants of the two phases with identification numbers so as to assure continuity and anonymity. We also gave every participant a small gift after completion in order to help increase response rate.

We collected 386 completed responses from 60 groups with three or more members out of 427 copies distributed among 67 groups in the first phase (response rate 90.40%) and 365 completed copies in 57 groups in the second phase out of the 386 participants and groups retained from the first phase (response rate 94.56%). After matching each participant with their two phase surveys and carefully checking their responses, we deleted respondents that were unable to match or had missed more than three items. This left us with a final valid sample of 339 participants in 54 groups (group size 3–13 with an average of 6.3 members). In our final sample, 64.9% of participants were men and 35.1% women, with an average age of 30.58 years (*SD* = 7.31), an education level of bachelor degree or above for 74.9% participants, and an average job tenure of 4.34 years (*SD* = 3.83).

### 3.2. Measures

To ensure the reliability and validity of measurements, we adopted well-established scales developed and used by previous researchers. All scale items in this study were presented in the Chinese language after a rigorous translation and back-translation process [36] from their English originals if needed. All questions but the demographic items used 5-point Likert scale with 5 for “strongly agree” and 1 for “strongly disagree”.

Group citizenship behavior. We adopted a 10-item scale developed by Chen, Lam, Naumann, and Schaubroeck [16] to assess group citizenship behavior. The scale was developed and validated in China and thus, has high content validity. A sample item is “My work group as a whole provides assistance to other work groups with heavy workloads” (α = 0.84, our adopted scale; the same below).

Constructive deviance. We measured constructive deviance with a 10-item scale developed by Galperin [4], which used a 7-point Likert scale originally. We adapted it to a 5-point scale to keep consistency among all our main measures since there are no remarkable differences between a 5-point scale and 7-point scale in terms of reliability and validity [37]. It includes two subscales—constructive deviance for the organization (a sample item is “I’ll deviate from organizational procedures to solve a customer’s problem”) and constructive deviance for the individual (a sample item is “To improve working procedures, I fail to follow my boss’s instructions”) (*α* = 0.89).

Environmental dynamism. We adapted a 3-item scale developed by De Hoogh, Den Hartog, and Koopman [38] to measure environmental dynamism. To emphasize the group context in our study, we adapted the original items in the scale. For example, we changed the item “Work environment is full of challenges” to “Group work environment is full of challenges” (*α* = 0.78).

Moral justification. We adapted a 4-item scale developed by Bandura, Barbaranelli, Caprara, and Pastorelli [39] to assess moral justification. In accordance with previous studies, we included the typical performance and social situation of constructive deviance in the measurement of moral justification [40]. For example, we changed the item “Violence for protecting friends is justifiable” in the original moral justification scale to “It is justifiable to break the company’s rules and regulations in order to improve work efficiency” (*α* = 0.76).

Group identification. We measured this control variable (see below) with a 4-item scale developed by Doosje, Ellemers, and Spears [41]. The scale consists of items on the cognitive, evaluative, and emotional aspects of group identity. A sample item is “I’m emotionally attached to my group” (*α* = 0.80).

Control variables. We controlled gender, age, education, and tenure at the individual level and the type and size of the company at the group level. We also controlled group identification at both individual and group levels for two reasons. First, although constructive deviance is a risky behavior, individuals with high group identification are more willing to take the risk with increased constructive deviance to improve organizational wellbeing [42]. Second, research has shown that identification with the licensing group is an important factor in moral credentialing effect [12].

## 4. Results

### 4.1. Data Aggregation and Validity Testing

Since our group level variables—group citizenship behavior, group identification, and environmental dynamism—were actually measured by individual employees’ responses and then, aggregated to the group level, we needed to assess the degree of group member consent on these group variables by calculating the Rwg value so as to justify the aggregation. The Rwg averages for group citizenship behavior, group identification, and environmental dynamism were 0.96, 0.93, and 0.91, respectively, all greater than the standard of 0.70. The result indicates that these three variables have sufficient intra-group consistency [43].

In addition, we conducted intra-class correlations (ICCs) to determine the reliability of group citizenship behavior, group identification, and environmental dynamism. We used *ICC_(1)_* to examine the degree of response variability at the individual level that is attributed to being part of the group. The three *ICC_(1)_* coefficients were 0.09, 0.40, and 0.29 for group citizenship behavior, environmental dynamism, and group identification, respectively. We used *ICC_(2)_* to examine the reliability of the group means. The three *ICC_(2)_* coefficients were 0.40, 0.81, and 0.71 for group citizenship behavior, environmental dynamism, and group identification, respectively. Generally speaking, *ICC_(1)_* is acceptable when it is beyond 0.12, while *ICC_(2)_* needs to be greater than 0.50 [43]. Although group citizenship behavior is not up to the best intra-group correlation level, the results of variance analysis show that group citizenship behavior in different groups reached significant differences, *F* (53, 285) = 1.97, *p* < 0.01. As pointed out by previous researchers [42], when group size is small, a significant inter-group difference can be used as an aggregation criterion in practice. Thus, all these results provide strong support for aggregating group citizenship behavior, group identification, and environmental dynamism from the individual level to the group level.

Next, we performed confirmatory factor analyses of the four main variables—group citizenship behavior, environmental dynamism, moral justification, and constructive deviance—with AMOS 19.0 (IBM Corp., Armonk, NY, USA) It is suggested that when the goodness of fit of the object model is better than that of the competing model, the discriminant validity of variables in the object model is considerably high [43]. As shown in Table 1, the confirmatory factor analysis results indicate that our hypothesized four-factor model (M4, i.e., group citizenship behavior, environmental dynamism, moral justification, and constructive deviance) was a better fit for the data (χ^2^/*df* = 1.475, *RMSEA* = 0.037, *CFI* = 0.955, *TLI* = 0.950) than any of the other three models (M3, M2, and M1). Meanwhile, we found in the confirmatory factor analysis results that the reliability of every scale is greater than 0.70, which means that the internal quality of the scales is good. Moreover, the composite reliabilities of all factors exceeded the required minimum of 0.70 [44]. With regards to the average variance extraction of each scale, not all of them have reached the ideal level, i.e., greater than 0.50; but composite reliability is all above 0.70, which implies that the convergence validity of each variable is still acceptable (as shown in Table 2) [44], with good construct reliability and adequate convergent validity. Furthermore, the square root of the average variance extraction of each variable is greater than the correlation coefficients among other variables, which demonstrates that the discriminant validity among the variables is quite high.

### 4.2. Descriptive Statistics and Common Method Bias Test

Table 3 shows the means, standard deviations, and correlations for all variables at both individual and group levels in the present study. As seen in the table, moral justification at the individual level has a significantly positive correlation to constructive deviance (*γ* = 0.55, *p <* 0.01); environmental dynamism at the individual level has a significantly positive correlation to both moral justification (*γ* = 0.19, *p <* 0.01) and constructive deviance (*γ* = 0.27, *p <* 0.01); and group identification at the individual level has a significantly positive correlation with constructive deviance (*γ* = 0.15, *p <* 0.01).

Common method bias could be an issue for all research studies like ours that collect all data from one single sample. Hence, following Zhou and Long’s [45] suggestion, we first conducted a varimax rotation analysis of principal factors for all variables in order to examine the presence and magnitude of common method variance. We found the first factor explained only 23.68% of the variance, that is, less than half of the variance (54.52%) explained by factors with eigenvalues greater than 1. Moreover, the independent variable (group citizenship behavior) and moderator (environmental dynamism) in our present study were both aggregated from the individual level and consequently, reduced the possible influence of common method variance. Results of the previous validity test also showed that the discriminant validity among the main variables was quite high. Therefore, we can reasonably conclude that common method variance in the present research was not significant.

### 4.3. Hypothesis Testing

We tested all hypotheses with multiple linear regression models in three steps. We first tested the zero model, then the moderating effect of environmental dynamism, and lastly, the mediating effect of moral justification.

Zero model. Sufficient inter-group variation in the dependent variable is a prerequisite for cross-level analysis. We built a zero model with moral justification and constructive deviance, respectively, before testing our main hypotheses. For moral justification, we found that the inter-group variation (τ00) and intra-group variation (σ2) were 20.66% and 6.36%, respectively, and the inter-group variation accounted for 76.47% of the total variation; for constructive deviance, the inter-group variation (τ00) and intra-group variation (σ2) were 12.64% and 16.01%, respectively, and the inter-group variation accounted for 55.88% of the total variation. These results on inter-group variation of moral justification and constructive deviance provided compelling evidence for the legitimacy of the cross-level analyses.

Moderating effect of environmental dynamism. We averaged both group citizenship behavior and environmental dynamism of all individuals in each group and aggregated the two variables to the group level. Then, we normalized each group’s group citizenship behavior and environmental dynamism results.

As shown in Table 4, M1 contains only individual- and group-level control variables and group citizenship behavior and environmental dynamism at the group level. Compared to M1, M2 adds the interaction between group citizenship behavior and environmental dynamism. The direct effect of group citizenship behavior on moral justification is insignificant (*γ* = 0.09, *p* > 0.05, M1), which is consistent with our hypothesis. However, the interaction between these two variables has a positive impact on moral justification (*γ* = 0.26, *p* < 0.001, M2). That is to say, group citizenship behavior is positively correlated with moral justification when environmental dynamism is high. 

In addition, as shown in Table 4, M3 consists of individual- and group-level control variables and group citizenship behavior and environmental dynamism at the group level, while M4 adds the interaction between group citizenship behavior and environmental dynamism to M3. The direct influence of group citizenship behavior on constructive deviance is insignificant (*γ* = 0.08, *p* > 0.05, M3) and consistent with our hypothesis. However, the interaction between group citizenship behavior and environmental dynamism has a positive effect on constructive deviance (*γ* = 0.15, *p* < 0.001, M4). This means that group citizenship behavior can significantly promote employee constructive deviance when environmental dynamism is high. Therefore, Hypothesis 1 is supported.

To better illustrate the moderating role of environmental dynamism, we drew Figure 2 and Figure 3 following Aiken and West’s [46] approach. Figure 2 suggests that when environmental dynamism is high, group citizenship behavior positively affects moral justification (*γ* = 0.20, *p* < 0.05); when environmental dynamism is low, however, the effect of group citizenship behavior on moral rationalization is insignificant (*γ* = −0.16, *p* > 0.05). By the same token, Figure 3 shows that when environmental dynamism is high, group citizenship behavior has a significantly positive effect on constructive deviance (*γ* = 0.19, *p* < 0.05); when environmental dynamism is low, however, group citizenship behavior has a significantly negative effect on constructive deviance (*γ* = −0.11, *p* > 0.05).

Mediated moderating effect. We tested Hypothesis 2 according to the steps suggested by Wen, Zhang, and Hou [47]. Hypothesis 2 proposes that moral justification mediates the relationship between the interaction of group citizenship behavior and environmental dynamism and constructive deviance. We have previously verified the direct effect of the interaction between group citizenship behavior and environmental dynamism on constructive deviance and moral justification, respectively. Thus, the first two conditions of the mediated moderating effect are met. Next, as shown in Table 4, when moral justification entered the model, its effect on constructive deviance is still significant but is marginal (*γ* = 0.26, *p* = 0.51, M5). Meanwhile, the impact of the interaction between group citizenship behavior and environmental dynamism on constructive deviance is no longer significant (*γ* = 0.004, *p* > 0.05, M5). That is to say, moral justification completely mediates the relationship between the interactive effect and constructive deviance. Thus, Hypothesis 2 is supported.

## 5. Discussion

Despite previous research findings that the constructive deviance seems rife with ethical concerns [7,8], it has rarely been conceptualized as an ethical decision issue [7,8], and little is known about how organizational factors, such as group citizenship behavior and environmental dynamism, systematically shape constructive deviance. Our study explores the process by which group citizenship behavior and environmental dynamism interact to influence constructive deviance. Our cross-level research results provide empirical evidence that the interaction between group citizenship behavior and environmental dynamism promotes employees’ constructive deviance through the mediation effect of moral justification. Now, we further discuss the theoretical and practical implications of these results, along with the limitations of the present work and the direction for future research.

### 5.1. Theoretical Implications

Our study makes several folds of theoretical contribution. As previously mentioned, constructive deviance is the result of an ethical trade-off between deontological and utilitarian approaches [7]. Yet, few empirical studies have explored the ethical challenges of constructive deviance [8]. In response to a call for such research, we treated constructive deviance as an outcome of an ethical decision and probed its antecedents and mechanism. Our findings explicitly address the ethical nature of constructive deviance [7,8] and expand its antecedents and mechanisms from an ethical decision-making perspective.

At the same time, our research also sheds a light on the specific conditions and the psychological mechanism through which group citizenship behavior is related to constructive deviance. Our research shows that under different levels of environmental dynamism, group citizenship behavior has opposite effects on constructive deviance because of different moral attributes of constructive deviance. Meanwhile, moral justification can fully explain the mechanism in the moral decision-making process of constructive deviance. Moreover, our research also responds to the appeal of Chen et al. [16] by clarifying the process by which group citizenship behavior influences individual behavior.

Our study also contributes to the understanding of moral licensing in the workplace. On the one hand, our work expands the sources of moral licensing. Previous research regarded employees’ good deeds (such as organizational citizenship and ethical leadership, etc.) as the only source of moral licensing in organizations [48,49]. Our findings suggest that group citizenship behavior can vicariously license constructive deviance by other group members. They responded to the call of previous researchers [15] about exploring the sources of moral licensing at the group level. Furthermore, we provide evidence of the mediating effect of moral justification as one route through which moral licensing occurs. Last but not least, our research empirically confirmed that attributional ambiguity is indeed a premise of vicarious moral licensing effect.

### 5.2. Practical Implications

As a matter of fact, constructive deviance has become a new normal for organizations today [5]. It dynamically seeks an optimal balance between maintaining stable work routines and altering them when environment changes [5]. That is, constructive deviance is a powerful practice to cope with the challenge of adaptive change. For example, Buchwald, Urbach, and Mähring [50] have suggested that people engaging in unofficial projects resulting from unsanctioned, bottom-up employee initiatives can bring about innovative ideas and solutions of potentially great benefit to organizations. Thus, constructive deviance becomes increasingly important in businesses today because it is advantageous to organizational change [50]. Based on our research findings, we would like to give a few suggestions for organizational leaders and managers on how to manage constructive deviance in practice.

It is the interaction between group citizenship behavior and environmental dynamism that positively influences constructive deviance. Therefore, on the one hand, managers should adopt a holistic, systematic approach to regulate employees’ behavior. They cannot directly encourage employees to engage in constructive deviance given its sensitivity and complications. However, they can indirectly promote employees’ constructive deviance through facilitating group citizenship behavior by increasing leadership support or building an equitable organizational climate [16]. On the other hand, some leadership styles, such as transformational leadership, can influence environmental dynamism by navigating external environments and raising employees’ sensitivity to environmental dynamism [51]. Thus, organizations should appoint proper leaders to stimulate group citizenship behavior and utilize the amplification effect of environmental dynamism in the moral licensing process. Besides, Reino, Kask, and Vadi [52] argued that the influence of organizational culture and the environment is bilateral. Combining these arguments with our findings, organizations can shape an appropriate organizational culture to enhance employees’ awareness of environmental dynamism, amplify the effect of environmental dynamism, and facilitate constructive deviance.

### 5.3. Limitations and Directions for Future Research

We would be remiss if we did not acknowledge a few limitations of our study that are worth addressing in future research. First of all, our variables are all self-reported, which could have resulted in common method bias. We used several methods to minimize its possible affect, such as ensuring participants of their responses’ confidentiality and collecting the data in two phases. We also adopted Harman’s single-factor test to detect common method bias. Results showed that common method bias was not likely to be a problem in our study. Furthermore, our cross-level study design could have somehow helped reduce the common method bias as well [43]. However, there is still a need to revalidate our data from multiple sources so as to obtain more robust results.

Second, we cannot exclude other possible theoretical explanations for our findings. One such explanation could be that group citizenship behavior and constructive deviance might concurrently occur because they share a common cause. For example, employees’ trust in their boss could be another reason that group citizenship behavior has an effect on constructive deviance. Findings of past studies have shown that subordinates’ trust in their leaders can promote the occurrence of group citizenship behavior [53] and constructive deviance [54], respectively. Considering the fact that these possibilities could contaminate our study, we adopted the two-phase approach to collect data and followed rigorous logical reasoning to assure the rationality of our model. Even so, future research should still control those alternative explanations more rigorously.

Third, our research tested our conceptual model only in service industries since the measure of constructive deviance that we used involved some customer service items. However, employees working in other industries such as manufacturing might engage in different types of constructive deviance from those of service employees [55]. Therefore, we encourage future research to develop or adapt a new scale to measure constructive deviance in general so as to generalize our findings. 

Last but not least, we suggest future research on constructive deviance take a multi-level or multi-subject perspective. Some researchers have pointed out that managerial constructive deviance simultaneously contains positive and negative components for employee behavior, so it is essential to explore its antecedents [54]. Meanwhile, it is argued that group citizenship behavior can facilitate group cohesion [32], which may further influence group constructive deviance behavior.

## 6. Conclusions

Drawing on moral licensing theory and social information processing theory, our study explores when and how group citizenship behavior promotes employees’ constructive deviance. Especially, it applies a multi-level approach to analyze the antecedents and conditions of constructive deviance and examine how moral justification acted as a mediator between the interaction of group citizenship behavior and environment dynamism and constructive deviance. Overall, our study not only enriches the management literature on constructive deviance and moral licensing theory, but also sheds light on how to manage constructive deviance in the workplace practically.

## Figures and Tables

**Figure 1 ijerph-17-08371-f001:**
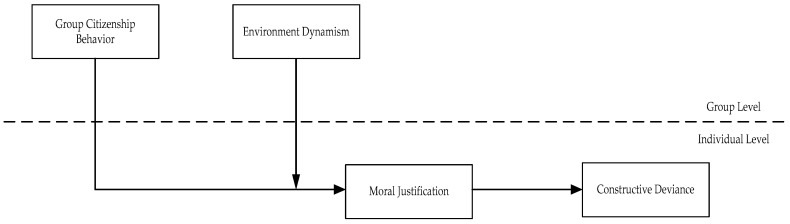
The research conceptual model.

**Figure 2 ijerph-17-08371-f002:**
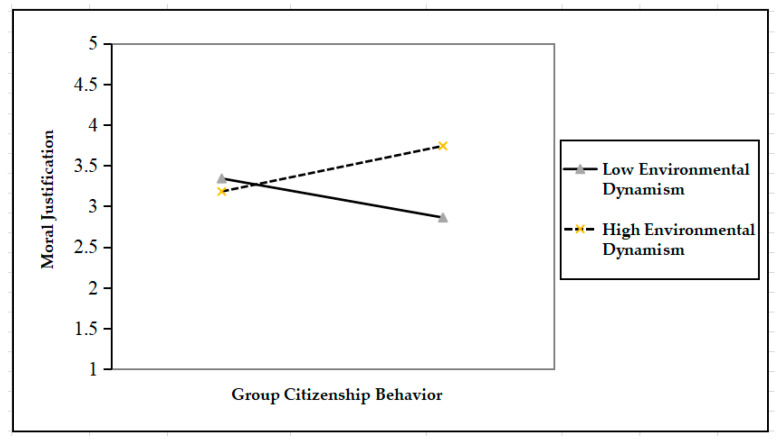
The moderating effect of environmental dynamism on the relationship between group citizenship behavior and moral justification.

**Figure 3 ijerph-17-08371-f003:**
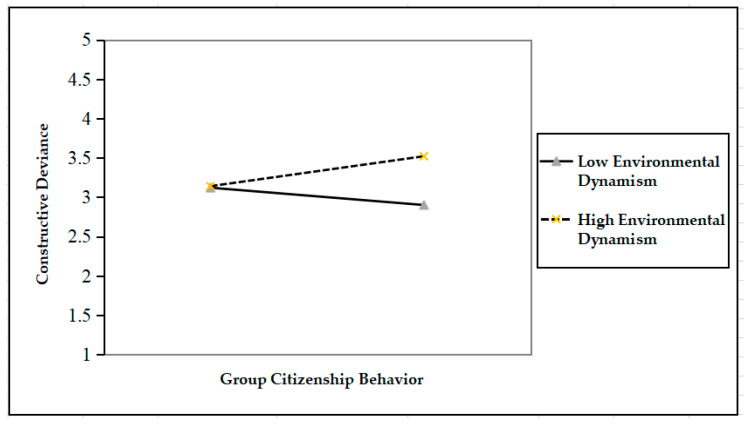
The moderating effect of environment dynamism on the relationship between group citizenship behavior and constructive deviance.

**Table 1 ijerph-17-08371-t001:** Confirmatory factor analysis results.

Model	χ^2^	df	χ^2^/df	CFI	TLI	RMSEA
M4	431.546	293	1.475	0.955	0.950	0.037
M3	754.382	296	2.549	0.850	0.835	0.068
M2	988.678	298	3.318	0.774	0.753	0.083
M1	1675.534	299	5.604	0.549	0.510	0.117

**Table 2 ijerph-17-08371-t002:** The composite reliability and average variance extracted.

Construct	Composite Reliability	Average Variance Extracted
Group citizenship behavior	0.85	0.39
Constructive deviance	0.89	0.45
Environmental dynamism	0.78	0.54
Moral justification	0.76	0.44

**Table 3 ijerph-17-08371-t003:** Descriptive statistics of variables.

Variable	M	SD	1	2	3	4	5	6	7
Ind. level									
1. Gender	0.35	0.48							
2. Age	30.58	7.31	−0.26 **						
3. Education	2.98	0.77	−0.36 **	0.11 *					
4.Tenure	4.34	3.83	−0.21 **	0.71 **	−0.01				
5. GI	3.68	0.66	0.09	−0.16 **	0.00	−0.19 **			
6. ED	3.69	0.75	0.08	−0.16 **	−0.18 **	−0.14 *	0.30 **		
7. MJ	3.28	0.52	0.12 *	−0.04	0.02	−0.14 *	0.10	0.19 **	
8. CD	3.29	0.53	0.05	−0.22 **	0.00	−0.24 **	0.15 **	0.27 **	00.55 **
Group level									
1. Size	11.46	4.28							
2. Type	1.31	0.47	0.02						
3. GI	3.68	0.44	−0.20	0.26					
4. GCB	3.91	0.28	0.10	0.14	0.53 **				
5. ED	3.68	0.56	0.09	0.30 *	0.29 *	0.52 **			

GI—group identification; ED—environmental dynamism; MJ—moral justification; CD—constructive deviance; GCB—group citizenship behavior. As for gender, men are coded as “1” and women as “2”. N (individual) = 339, N (group) = 54. ** *p* < 0.01, * *p* < 0.05.

**Table 4 ijerph-17-08371-t004:** Main effect and moderating effect.

Variable	MJ		CD		
	M1	M2	M3	M4	M5
Intercept	3.30 ***	3.17 ***	3.30 ***	3.23 ***	3.29 ***
Individual level					
Gender	−0.05	−0.05	−0.21 ***	−0.21 ***	−0.20 ***
Age	0.02	0.02	−0.08 **	−0.08 **	−0.09 **
Education	−0.02	−0.02	0.05 ^+^	0.05 ^+^	0.05 ^+^
Tenure	0.01	0.00	0.00	0.00	0.00
ED	−0.06	−0.06	0.01	0.01	0.03
TI	−0.06 **	−0.06 **	−0.04	−0.04	−0.02
MJ					0.26 ^+^
Group level					
Type	−0.03	−0.09	0.07	0.04	0.09
Size	−0.03 ^+^	−0.01	−0.01	−0.01	0.00
GCB	0.09	0.02	0.08	0.04	0.03
ED	0.13	0.18 **	0.13 *	0.16 **	0.06
TI	0.03	0.06	0.05	0.06	0.03
Interaction					
GCB × ED		0.26 ***		0.15 ***	0.01
Variance decomposition					
Intra-group variance	60.05%	60.04%	110.60%	110.60%	110.24%
Inter-group variance	160.99%	100.83%	110.94%	100.23%	60.94%

TI—group identification; ED—environmental dynamism; MJ—moral justification; CD—constructive deviance; GCB—group citizenship behavior. As for gender, men are coded as “1” and women as “2”. N (Individual level) = 339, N (Group level) = 54. *** *p <* 0.001, ** *p <* 0.01, * *p <* 0.05, ^+^ 0.05 < *p <* 0.10.

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
