# Peer review of "The Paradox of Group Citizenship and Constructive Deviance: A Resolution of Environmental Dynamism and Moral Justification"

_ijerph, 2020, doi:10.3390/ijerph17228371_

Round 1
Reviewer 1 Report
I understand that it is the discretion of the editors to accept this paper, but in my opinion, the results will establish a new precedent baseline value for AVE, which is not good for the IJERPH’s reputation if noticed by critics in the field, and could create a questionable image as to what basis such results were allowed to be acceptable, without any comparable explanation by authors or without justifiable backing by statistical references.
There are multiple theories referenced toward, yet the hypotheses have been built on insufficient literature in the first place, but the fact they refuse to elaborate the discussion for the readers to understand the logic, is a disappointing situation.
In addition, the paper needs a lot of work for three reasons,
- The authors state and refer to some past studies in esteemed journals of high rank, that those studies did not provide these AVE statistics. This does not mean those studies’ AVE values were below 0.40. It was a mere oversight during the review process and the relevant editors’ decision to not demand such statistics from those authors. It is not a justification for lower AVE values.
- The authors must have tried to establish in the discussion that they are aware that these results have statistical discrepancies, in detail write those discrepancies that what’s the standard requirement and how their results are slightly inferior inn reliability. Yet, have an overall meaningful interpretation. There is no intention visible on the author’s part with regards to this aspect.
- The literature and hypotheses building along with the discussion section all need to be knitted well by describing how the leadership roles play between the relationships pf the variables of this model. The authors have left too much on the reader’s imagination to establish the true relationships between variables.
Taking a holistic view, to be frank, I am disappointed with the authors' efforts. They have spent ample time on efforts to justify the flaws and discrepancies in their paper, but have not tried to elaborate for the readers how this paper is lacking in certain areas and care must be taken by other researchers in using the results if this study for future reference. For example, smartpls is a robust software if the sample size is small, but most researchers in the initial years of its launch justified using smartpls instead of SPSS because of this reason, which was not enough and then HAIR et. al. later pointed out this malpractice in their subsequent paper.
Based on these observations, in my humble opinion, there is no merit in publishing this paper. I do not wish to accept the verbose arguments of the authors, it would have been more productive of they spent their time on revising the paper rather than telling the reviewers that they are “reluctant to let our paper distracted by other plausible ideas”, the suggestion was in fact related to the parts of the paper which the authors claim but have not sufficiently described, the suggestion was not at all something unrelated to the model.
So, my recommendation is to reject the paper.
Author Response
Thank you for your time.
We'll respect the academic editor's final decision.
Have a great day.
Reviewer 2 Report
Dear Authors,
I really appreciated to review the paper, as indeed you incorporated the suggested improvements, namely further details of the participants and their selection, so as, explaining the way of how managers can promote constructive deviance.
Now, the Abstract, Introduction and Conclusions are linked and clearly detailed.
Please find the reviewed document, with yellow highligjts and comments.
I suggest to have a final text editing review.

Author Response
We five authors highly appreciated your time and collegiality and learned a lot from your candid but helpful comments and suggestions.
Thank you very much for your great help.
Reviewer 3 Report
The authors addressed all my concerns; therefore I support publication without other changes.
Author Response

(The authors gave the same response as above.)

Reviewer 4 Report
- The research gap should explain in the introduction. Why you chose group citizenship behavior in your model is not clear.
- What is the unique characteristics of service companies rather than other industries is needed to address.
- During hypotheses development, you need to combine the theory and your argument, then develop the hypotheses.
- The theoretical contribution should address in the conclusion, and the model seems too simple with two theories.
- I suggest that you adopt appropriate academic expression. In general, the writing of the paper could be improved.
Author Response
We five authors highly appreciated your time and collegiality and learned a lot from your candid but helpful comments and suggestions.
Attached please find our detailed responses to your feedback.
Thank you very much for your great help.

This manuscript is a resubmission of an earlier submission. The following is a list of the peer review reports and author responses from that submission.
Round 1
Reviewer 1 Report
This is a very interesting study from a practical perspective as well as addition to existing literature. I have the following suggestions, which the authors may take into consideration with the aim to improve the paper. 1. Lines 63-64, “Since the pro-organization motivation of constructive deviance is implicit, it could be misconstrued as ethical violation unless its true drive can be detected by others.” , please cite a reference, or rephrase it as an opinion. 2. Lines 67-71 “Particularly, when employees are in … promotes constructive deviance.”, needs citation. In the introduction section, while you are not comparing arguments of previous research, this must be clarified as an opinion or an argument… for example, “it could be argued that when employees are in…….”. Very similar to how you state your argument in line 73. 3. Hypothesis 1 is unorthodox in nature. While it posits relationship of IV with DV, there is actually no direct path modeled in this study. It is suggested that authors need to split the hypothesis into two sections (1a & 1b), or perhaps (at least) two separate sub-headings for clarifying the relationships between IV and DV independently from the discussion on the role of moderator for its probable influence on this relationship. 4. Constructive deviance under Hypothesis 1 only focuses on individual level behavior. It would make the discussion stronger if authors add how it relates to or formulates into group level deviance, it would make more sense if a leader or group level deviance is added to the arguments. 5. Although it is not a strictly express rule, a moderated mediation hypothesis usually follows the mediation hypothesis, and not prior to mediation hypothesis. 6. Hypothesis 2 is too weak. Considering that mediation needs to be justified for relationship with independent variable as well as dependent variable, there is merely one paragraph that defines moral justification, and then another mediocre paragraph that weakly argues about how a logic is being applied in this study’s context to establish mediation. Specific citation to previous works on dependent variable or similar variables. If no studies on exact relationship between group OCB and moral justification are found, then there must be a previous study on moral justification and constructive deviance. (if no study exists, then mediation hypothesis is fundamentally flawed). a mediation hypothesis can not be established otherwise 7. Lines 175 0 180 “This reframing and elimination process is the mediation effect of moral justification. In other words, when environmental dynamism is high and … organizational rules or norms, such behavior cannot be justified and forgiven.”. Need a citation to support this line of argument. 8. While the study presents implications for the leadership (line 404-405) and managers, there is almost no mention of leadership or managerial roles in the introduction or the hypotheses development section. The whole introduction section, as well as the hypotheses needs to present a well-knitted background on at least one leadership style (e.g. ethical leadership) for its influence on group behaviors, or perhaps discussion on positive leadership styles as a whole. This is necessary because the authors posit this study as multi-level, OCB must be given a contextual background with respect to how leaders can influence this individually as well as at the group level. Specifically, environmental dynamism is also a macro-level aspect which is bred by leadership and top management. Moreover, the authors use MLT which assesses individual behavior, so bridging the individualistic behavior onto group behavior needs to explained. This can be addressed by discussing leadership. 9. The methods section should add an explanatory statement why two-wave data collection was done to tackle CMV (e.g. avoiding social desirability by respondents while filling questionnaire to makes responses more congenial for IV and DV, etc.) 10. The measure constructive deviance by Galperin (2012) was developed with 7-point likert scale, please identify the reason in your study as to why you have used 5-point instead of 7 point scale, 11. Line 223, “we slightly revised the original items in the scale.”, it is suggested to use the were adapted instead of revised, a “revised” scale would suggest that the psychometric properties may need to be reported again from instrument development perspective, which is not the case here. 12. Line 259, “all greater than the standard of .70.”, citation needed 13. Although results briefly refer to model fit, the details on reliability and validity of the measures is not given, please specify whether the Cronbach’s Alpha given under “3.2. measures” is that of the established measures from past studies or of this study? The authors have not reported construct-level Composite Reliability and Average Variance Extracted, which is usually preferred. 14. The discussion should supplement a bit more on practical aspects. add to how the output variable is related to various organizational situations, how leaders should identify it and how it should be perused in terms of sales growth and performance evaluation of employees. How is the deviant behavior informed to other employees as a preferred behavior as a cycle between OCB and influence at the individual level?Author Response
Please see the attached document. Thanks.

Reviewer 2 Report
Dear authors,
I really appreciated to read your paper.
I suggest to include missing relevant information in the Abstract, as just in 3. Method, more details are explained.
Information regarding the nationality of the participants, so as the selection os five service companies should be described in a very brief and short way in the Abstract.
This information is really crucial to provide from the beginning deeper knowledge.
Furthermore, please explain, under point 5.2. (Practical Implications), how managers can promote constructive deviance.
Reviewer 3 Report
Moving from the perspective of ethical decision making and using the theorical frameworks of moral licensing theory and social information process theory, the authors conducted a study on group citizenship behavior as a variable that facilitates constructive deviance, emphasizing the mediating role of moral justification and the buffering effect of environmental dynamism. 339 workers in 54 groups were involved in a survey study, and results showed that the moral justification fully mediates the relationship between the group citizenship behavior and environmental dynamism interaction (at the group level) and constructive deviance (at the individual level).
Overall, the paper is well written and referenced, and the data analysis strategy used is correct. In my opinion, authors manage the statistical tools well. Discussions are more than adequate and appropriately detailed and in line with results obtained, and so are the limitations. This paper may give an important contribution to the topic of constructive deviance. Hence, my review points out only few aspects that could be taken into account to improve the manuscript.
1. Although the construct of “environmental dynamism” obviously relates primarily the organizational level, the authors treat it at the group level. I believe that it would be useful to argue this aspect also in the part of the development of the hypotheses, highlighting literature supporting this option.
2. The use of the group identification as control variable could somehow be argued. why is it important to check the results for this variable? how could it influence the hypothesized relationships?
3. Environmental dynamism should be considered in relation to the organizational culture and the values that could promote it. This could be addressed in the context of discussion and further research developments
Minor issues
1. The statistics are not consistently reported in line with any style familiar to me (like APA, or Harvard guidelines); e.g., the 0 before the decimal point. I’m not sure this follows the journal guidelines. Please, check this.
2. Please add choices range of the Likert scales used for each measure
I would like to thank authors to give me the opportunity to read this research.
Round 2
Reviewer 1 Report
ROUND 2
- With reference to Comment 13, and subsequently reported values of Average Variance Extracted by the authors, this study’s reported level of OCB construct’s convergent validity (much lower than 0.5 benchmark) is an unusually low level and fundamentally challenges the prevailing practices.
- Convergent validity means that an instrument’s items sufficiently represent what is being measured, and they converge into a single construct. Latest SEM techniques and guidelines (Hair et al., 2017; Hair et al., 2013; Sarstedt et al., 2011)recommend that AVE value of 0.50 means that item loadings making up the construct are 0.7 on average. Irrespective of the software application used for data analysis, the fundamental principles of SEM technique remain the same.
Hair, J., Hollingsworth, C. L., Randolph, A. B., & Chong, A. Y. L. (2017). An updated and expanded assessment of PLS-SEM in information systems research [Article]. Industrial Management & Data Systems, 117(3), 442-458. https://doi.org/10.1108/imds-04-2016-0130
Hair, J., Ringle, C., & Sarstedt, M. (2013). Partial Least Squares Structural Equation Modeling: Rigorous Applications, Better Results and Higher Acceptance. Long Range Planning, 46(1-2), 1-12. https://doi.org/10.1016/j.lrp.2013.01.001
Sarstedt, M., Ringle, C. M., & Hair, J. F. (2011). PLS-SEM: Indeed a Silver Bullet. The Journal of Marketing Theory and Practice, 19(2), 139-152. https://doi.org/10.2753/mtp1069-6679190202
- Although “exploratory” studies show lower level of AVE is acceptableduring dimension reduction to develop instruments, most of them identify a value of at least 0.40 as bare minimum. Moreover, such practice of accepting lower AVE may be noticed to be not prevalent for studies using CFA for established instruments, similar to the currently reviewed study.
- The most important aspect is that it sets a precedent. For an instrument’s validity, such a citable precedent is questionable (if this study’s reported AVE becomes part of the extant literature). Such methodological and psychometric properties have taken decades of research by renowned scholars to establish benchmarks for AVE and should rather be based on statistical inferences and not one reference (the authors cite Chen, Xu and Fan (2012) which I could not find in my attempt to search for the book to look up the AVE benchmark indication in it)
- It is strongly suggested actual research studies may be cited instead of a book to support lower AVE acceptability. Citing outdated practices is not a justification to use lower AVE values. The following references be also added in the section reporting AVE values.
Fornell, & Larcker, D. F. (1981). Structural equation models with unobservable variables and measurement error: Algebra and statistics. J Mar Res, 18(3), 382-388. https://doi.org/10.2307/3150980
Lam, L. W. (2012). Impact of competitiveness on sales people's commitment and performance. Journal of Business Research, 65(9), 1328-1334.
- Authors must also report variance inflation factor (VIF) in order to show multicollinearity was not a problem in the dataset. VIF values greater than 3 would suggest serious problems.
Hair, J. F., Black, W. C., Babin, B. J., & Anderson, R. E. (2010). Multivariate data analysis (7 ed.). Upper Saddle River, NJ, USA: Prentice-Hall, Inc.
- In my humble opinion, a limitation must clearly be stated in the limitations section of the study that “the AVE values in this study were below the suggested benchmarks and further studies may be needed to validate the instrument in Chinese context and other researchers may take caution in generalizing the findings of this research”.
- With reference to comment 8 from round 1 of review, please add the leadership context in the hypotheses development section. Since the authors have referred to transformational leadership style in the discussion, they may proceed to define this leadership style in the hypothesis development section and refer to a few of past studies on positive outcomes of this leadership style (preferably with outcome variables representing group behavior).
- With reference to reviewer’s comment # 14 in previous review stage, please add some discussion on:
- how the output variable is related to various organizational situations, ADDITIONS HAVE ALREADY BEEN MADE BY AUTHORS
- how leaders should identify and estimate existence of constructive deviance NOT SPECIFIED BY AUTHORS
- how it should be perused (viewed and rewarded) in terms of sales growth and performance evaluation of employees. NOT IDENTIFIED BY AUTHORS IN THEIR DISCUSSION
- How is the deviant behavior informed to other employees as a preferred behavior as a cycle between OCB and influence at the individual level? NOT IDENTIFIED BY AUTHORS IN THEIR DISCUSSION